# A brighter future? Stable and growing sea turtle populations in the Republic of Maldives

**Jillian A. Hudgins**[1]*, **Emma J. Hudgins**[2], **Stephanie Köhnk**[1], **Enas Mohamed Riyad**[3], **Martin R. Stelfox**[1]

**1** Olive Ridley Project, Sabden, United Kingdom, **2** Department of Biology, Carleton University, Ottawa, Canada, **3** Environmental Protection Agency, Malé, Republic of Maldives

* jillian@oliveridleyproject.org

**Data Availability Statement:** The data are available at DOI: 10.5281/zenodo.7772644 (https://doi.org/10.5281/zenodo.7772644)

## Abstract

The Indian Ocean represents a significant data gap in the evaluation of sea turtle population status and trends. Like many small island states, the Republic of Maldives has limited baseline data, capacity and resources to gather information on sea turtle abundance, distribution and trends to evaluate their conservation status. We applied a Robust Design methodology to convert opportunistic photographic identification records into estimates of abundance and key demographic parameters for hawksbill sea turtles (*Eretmochelys imbricata*) and green sea turtles (*Chelonia mydas*) in the Republic of Maldives. Photographs were collected *ad hoc* by marine biologists and citizen scientists around the country from May 2016 to November 2019. Across 10 sites in four atolls, we identified 325 unique hawksbill turtles and 291 unique green turtles—where most were juveniles. Our analyses suggest that, even when controlling for survey effort and detectability dynamics, the populations of both species are stable and/or increasing in the short term at many reefs in the Maldives and the country appears to provide excellent habitat for recruiting juvenile turtles of both species. Our results represent one of the first empirical estimations of sea turtle population trends that account for detectability. This approach provides a cost-effective way for small island states in the Global South to evaluate threats to wildlife while accounting for biases inherent in community science data.

## Introduction

The importance of conserving sea turtles in the Indian Ocean region has been recognized by the signatories to the Indian Ocean South East Asian Marine Turtle Memorandum of Understanding and by policies implemented by countries in the region (see for example [1–3]). However, to guide conservation policy, assessments of abundance, distribution, and temporal trends (particularly for apparent survival and recruitment) are necessary to understand management units and inform conservation action [4, 5]. Obtaining such information can be both expensive and logistically challenging, particularly for nations with limited capacity and resources. Both the green and hawksbill turtle are listed on the International Union for Conservation of Nature (IUCN) Red List as endangered [6] and critically endangered [7],

**Funding:** The authors received no specific funding for this work.

**Competing interests:** The authors have declared that no competing interests exist.

respectively. However, evaluation of the conservation status of nesting and foraging green and hawksbill sea turtle populations in the Republic of Maldives has been hindered by limited baseline data (but see [8, 9] for recent studies). Previous studies of long-lived organisms have shown that their life-history traits, which coevolve with long life, result in populations having a limited capacity to respond to rapid change [10, 11]. This is concerning because, as anthropogenic disturbances and impacts accelerate, sea turtles may not have the ability to adapt, particularly when range expansion is limited due to unsuitable nesting sites [12].

In marine turtles, the use of photographic identification (photo-ID) has proliferated in recent years [13–17], opening new opportunities to study these animals in their natural environments while also minimizing disturbance [18]. Historically, capture-mark-recapture studies on sea turtles used flipper tags or passive integrated transponder (PIT) tags to study females at nesting sites [19]. Few studies have focused on adult males, juveniles, or sub-adult turtles. The lack of information on large parts of sea turtles' life cycles is now worrying, due to the concerns regarding significant anthropogenic threats to sea turtles in all life stages and habitats, such as habitat loss, bycatch, and entanglement in ghost gear [20]. Specifically, the Indian Ocean has been listed as having the greatest data needs upon which to base assessments [20].

We applied a Robust Design methodology to convert opportunistic photo-ID records from the Maldives into estimates of sea turtle abundance and key demographic parameters. When appropriately specified, Robust Design estimates are more accurate and precise than those obtained through the application of closed and open models in sequence, because they allow for an estimation of apparent survival and abundance while accounting for temporary emigration [21]. They are also able to take into account differences in detectability of individuals within the population and across sampling periods, a phenomenon present in most opportunistic survey strategies [21, 22]. We relied on the suite of potential population models formulated by [22] in RMARK to determine the demographic processes that best described the turtle populations at each of our sites. From there, we extracted the relevant demographic parameters and population estimates across time in order to compare abundance trends, stability, and abundance of turtles across species and sites.

In this case study, we show how this photo-ID project is contributing estimates of key demographic parameters for hawksbill and green sea turtles, and how this information can be valuable in locations with limited capacity to monitor species at risk. We anticipated that sites within the same atoll would show similar seasonal fluctuations in abundance, and greater stability and abundance of hawksbills relative to green turtles.

## Methods

### Study site

The Republic of Maldives is located in the central Indian Ocean. It is an archipelago of ~1,200 islands spread over an area greater than 90,000 km$^2$ [23] including 4,500 km$^2$ of reef [24], which serves as important habitat for sea turtles. As of 2021, there are 79 protected areas, including three UNESCO Biosphere Reserves within the country. Of the seven species of sea turtles, five have been recorded in the Maldives: the green turtle (*Chelonia mydas*), hawksbill turtle (*Eretmochelys imbricata*), olive ridley turtle (*Lepidochelys olivacea*), loggerhead turtle (*Caretta caretta*), and leatherback turtle (*Dermochelys coriacea*). There are two monsoon seasons in the Maldives: "*iruvai*": the northeast monsoon or dry season (represented here by the period November to April) and "*hulan'gu*" the southwest monsoon or wet season (represented here by the May to October period) [25]. The dry season is the high season for tourism in the Maldives. Additionally, water is clearer and calmer, making turtles easier to spot. Generally, visibility is lower during the wet season and there are fewer tourists in the water. We expected

these factors to cause seasonal variation in turtle detectability, which we sought to capture and separate from seasonal variation in demographic parameters.

Green sea turtles are listed as endangered with a stable or decreasing trend for the North-West Indian Ocean Regional Management Unit (NWIO RMU) [26]. The national Red List of the Maldives also lists the species as endangered, based on a significant decrease in documented nesting activity since the 1980s (9). A similar finding for hawksbill turtles with an even more pronounced decrease in recorded nesting led to the national listing of this turtle species as critically endangered in the Maldives [8]. In contrast, the NWIO RMU population for hawksbills is listed as increasing and the Maldives is thought to be one of the most important foraging areas for hawksbills in the Indian Ocean [26]. As such, monitoring sea turtle populations within the NWIO RMU could further validate the key importance of this region for the persistence of these imperiled species.

Previous studies from other regions have revealed the high site fidelity of both species [17, 27–31]. Given their consistent interactions with particular locations, resident individuals may be particularly vulnerable to anthropogenic threats, such as habitat degradation, marine debris, poaching, and entanglement at these sites. Sea turtles, their eggs, and their habitats were legally protected in the Maldives in 2016 by the Environmental Protection and Preservation Action (Number 4/93 Section A—Conservation of Biodiversity). Despite this, major threats to sea turtles in the Republic of Maldives currently include egg and meat poaching, the pet trade, loss of nesting habitat, and entanglement in marine debris [32, 33]. A report by the Maldives Marine Research Centre [34] noted that awareness of turtle conservation issues remains limited among the Maldivian public, and recommended increased enforcement and stricter penalties for violations to conservation regulations. Enforcement in tight-knit, low density, and highly dispersed communities remains challenging, particularly in remote atolls. Fisheries in the Maldives pose little threat to turtles, as the primary method of fishing is live bait pole-and-line, which has low turtle bycatch [35]. Tourism and trade-related exploitation of sea turtles and their habitats coupled with an increasing human population are creating an uncertain future for these animals in the Maldives. However, involving tourists and locals in a citizen science data collection project such as this one can build both public awareness and have policy implications (see [36] for policy outcomes of a previous study).

## Data collection & processing

The Olive Ridley Project (ORP) began collecting new and historical photographs of foraging and nesting turtles in the Maldives in 2014. ORP's photo-ID project aims to help fill the gaps in scientific knowledge, increase the certainty in population demographic estimates, and aid in the creation of a consistent, long-term dataset to inform government policy and management at a local and national scale. As of late 2019, the dataset included almost 25,000 sightings of sea turtles across the country. Photos have been collected from sites spanning ~46% of the country by over 350 individual submitters. Thanks to a network of citizen scientists and biologists spread across the country, this program has been extremely cost effective. Data submitters provided photos for identification, survey site name and GPS coordinates, date, and an estimate of straight carapace length for each sighted turtle. Size estimates of turtles were only encouraged when they could be approached within 5 m. Adult male turtles were identified based on the length of the tail [37]. All turtles of 60 cm carapace length or below were labeled "juvenile", those without a size estimate were recorded as life stage unknown, and adult sized turtles without clear identification as males based on tail length were recorded as sex unknown. As there was no way to verify the size that a data collector submitted, it is possible that female turtles may have been incorrectly identified as juveniles and vice versa. Before data collection, all data

submitters received training information, which included behaviour and data standards in the form of a virtual or in-person presentation and detailed written instructions. From 2018 onward, data submitters additionally received a Code of Conduct infographic [38] that outlined best practice for collecting photographic data. This ensured maximum data quality and minimum impact to the animals.

We used Capture-Mark-Recapture (CMR) methods to analyze photo-ID data. Sea turtles can be individually identified based on their arrangement of facial scales [16]. These markings are stable over the long term, allowing non-invasive CMR methods. When CMR is adapted to photo-ID data, a photograph is used as the "capture", the pattern of scales is the "mark", and a subsequent photo is the "recapture".

Data collection involved no handling of animals. Data collection and analysis were done based on research permits issued by the Maldivian Environmental Protection Agency and the Ministry of Fisheries and Agriculture (permit numbers EPA/2018/PSR-T01, EPA/2019/PSR-T06, (OTHR)30-D/INDIV/2019/585). In addition to long-term monitoring sites across the country, data were collected *ad hoc* from marine biologists and dive guides stationed in resorts, guest houses, and on dive boats, as well as from locals and tourists. Data collected *ad hoc* are often highly variable in quality and completeness. We found large differences in survey effort around the country, and high turnover of staff at many locations, leading to data gaps and lack of data continuity. These large differences in effort resulted in the restriction of sites and time periods examined within our models (see *Population Models*).

When a turtle was sighted, it was approached slowly and as many photographs as possible were taken of the animal, aiming to capture at least one photograph of each side of the face and one of the carapace. The date and location were recorded, as well as turtle life stage and sex, when possible. Photos that were blurry or did not clearly show the turtle's face were discarded and incomplete submissions (missing dates or sites) were not entered into the dataset. Photos containing multiple animals were duplicated and cropped to show only one animal each. Matches were confirmed by eye. The database was compiled by one person, the lead author, but a minimum 50% subsample of the entire database was checked by two independent trained verifiers via visual inspection. Turtles were given a unique ID code. Photographs that could not be matched to an existing individual in the database were assigned a new identification number. Photographs of at least the right side of the head (but ideally both sides of the face) were required to assign a new identification number. Photographs of only the left side of the head that could not be matched to an existing individual remained in the database for future identification but were not assigned a new identification number. Incomplete data (missing identification, dates, or sites) were not entered into the dataset. The dataset was migrated to a global online platform in late 2019 (Internet of Turtles: iot.wildbook.org), which will allow future matches to be confirmed using the Wildbook Image Analysis machine learning and computer vision stack in addition to visual confirmation.

## Population models

CMR data are often used to model animal populations as either open (movement in and out of the population) or closed (no movement in or out of the population) [39, 40]. Pollock's Robust Design (RD) offers an intermediate approach, and has been used to estimate abundance and demographic patterns of other taxa such as Asian elephants (*Elephas maximus*) [41], tigers (*Panthera tigris*) [42], and several species of dolphin [22, 43]. Notably, RD can accommodate diverse patterns of site association, such as temporary emigration, and differences in the rate of immigration to a site and emigration out of it, known as Markovian emigration [21, 44, 45]. Markovian emigration may be an early warning sign of population decline and it signifies that

it is important to keep turtles on their home reefs and reduce disturbances. RD relies on open sampling occasions known as "primary periods", within which are a number of closed sampling occasions, known as "secondary periods" [21]. Closure is assumed within primary periods but not between them.

In this study, six months was chosen as the primary period and one month for the secondary period. Two six-month periods aligning with the country's monsoon seasons to test seasonality were defined: May to October and November to April. The RD structure thus assumed that the population was closed to immigration and emigration across all months within a six-month timespan (monsoon season) but open between monsoon seasons. Additional RD protocol assumptions are as follows: 1) individuals are correctly identified; 2) there are no changes in demographic parameters within the secondary period; 3) capture and apparent survival probability do not vary among individuals.

We considered separate datasets for green and hawksbill turtles. For each dataset, we built a capture matrix (i.e., a binary table with individuals in rows and secondary periods in columns). A 1 was entered if an individual was detected during a sampling period, regardless of how many times it was detected, and a 0 was entered if the individual was not detected. These models were input into the RMark program. A set of 30 models composed of parameters for abundance, apparent survival rate, temporary migration rate, and capture probability were fitted to the data with program R, package RMark [46], to construct models from program MARK [47]. Our analysis tested 30 different models in order to explore the different ways that capture probability, apparent survival, temporary emigration, and immigration have changed over time. It is important to note that this was an exploratory analysis where we were exploring competing models and not testing any hypotheses, similar to [22].

Although the photo-ID database for the Maldives spans almost half of the country, for this study, we restricted site-level analyses to ten reefs across four atolls (see S1 Table, S1 Fig for details) given the lower number of total turtles and inconsistent survey effort at the remaining atolls. As the RD analysis performs better with constant detection effort, only sites with consistent data collection from May 2016 to November 2019 and with at least 10 identified individuals were analyzed. In some cases, the first six-month interval was omitted due to paucity of data. Although longer-term data sets are needed to fully understand population stability and trends, shorter-term photo-ID datasets such as this one can be used to understand some demographic parameters.

At each reef, we extracted population estimates, population growth rates, and population coefficients of variation. For the second to the last six-month closed intervals, the relative change in abundance was calculated by dividing the present population estimate by the previous population estimate. The average annual population growth rate for the entire modelled period was calculated as

average annual population growth = ((final population estimate /initial population estimate) $^{(1/T)}$ -1)*100%

where T is the total length of the time series (3.5 years in most cases). The coefficient of variation was calculated as the standard deviation of the population growth rates calculated for each of the seven six-month intervals divided by their mean. The coefficient of variation is a measure of the variability in population growth, and hence in the stability of populations. Unstable populations may require greater conservation investment because instability often precedes extirpation [48].

One more complex aspect relating to Robust Design methodology is the need to estimate and correct for the impact of overdispersion on model results. This involves performing four tests for homogeneous detection and apparent survival probabilities across turtles, both of which can lead to overdispersion in the resulting *RMark* model. If the model fails any of these

tests, we calculated a variation inflation factor that was used to calculate a penalized AIC score (QAIC) for model selection, which leads to appropriate model selection inferences [49]. In practice, this means penalizing more complex models in favor of simpler models when assumptions of consistent apparent survival and detection are violated. We performed these tests using the *R2ucare* package on data from each site pooled by primary interval [50]. Because of the high rate of non-detection in primary intervals, we also had to pool individual turtles into groups of three in order to perform these tests.

## Results

In total, between 2016 and 2019, 325 unique hawksbill turtles (henceforth, hawksbills) and 291 unique green turtles (henceforth, greens) were identified.

Overall, out of the 325 identified hawksbills, 164 (50%) were sighted for only one year, 66 (20%) were sighted during two separate years, 50 (15%) were sighted during three separate years and 45 (14%) were sighted all four years. Each year, between 62–86% of all sighted hawksbills were sighted between November and April (the dry season, mean = 71%, sd = 11%), and 14–38% were sighted between May and October (the wet season, mean = 29%, sd = 11%). Across years and at the level of atolls, 23–33% of hawksbills were sighted at Ari (mean = 28%, sd = 4%), 11–15% were sighted at Baa (mean = 13%, sd = 2%), 34–39% were sighted at Kaafu (mean = 37%, sd = 2%), and 12–30% were sighted at Laamu (mean = 21%, sd = 8%).

Out of the 291 identified greens, 131 (45%) were sighted in only one year, 72 (25%) were sighted during two separate years, 62 (21%) were sighted during three separate years, and 26 (9%) were sighted in all four years. Each year, 31–82% of all sighted greens were sighted between November and April (the dry season, mean = 62%, sd = 22%), and 18–69% were sighted between May and October (the wet season, mean = 38%, sd = 22%). Across years and at the level of atolls, 34–77% of all greens were sighted at Laamu (mean = 50%, sd = 19%), and 23–66% of greens were sighted at Lhaviyani (mean = 50%, sd = 19%). No turtles of either species were photographed in more than one atoll.

### Life stages

Juvenile turtles made up the majority of the photographed individuals at all reefs, except for at Lh.Caves, where adults made up 52% of the greens photographed (n = 30) (Fig 1). There were far more adult greens photographed (n = 96) compared to adult hawksbills (n = 13). No more than two adult male hawksbills or two adult female hawksbills were recorded at any reef in this study, though there was a large proportion of individuals at some reefs whose sex and size were not recorded by the observer. The adult sex ratio was 1M:1.9F across all analyzed reefs for greens, and 1M:7.2F in hawksbills. Individual ratios at each site varied drastically between reefs, from only one of the sexes being identified in adults (greens: L.HithadhooW; hawksbills: A.Dhidhdhoo, B.Dhonfanu, L.Olhuvelhi) to female biased ratios of 1:3.5 for greens (L.Olhuvelhi) and 1:1 for hawksbills (K.BHHR, K.BHTR, L.Olhuveli).

### Population models

Out of the six reefs studied for hawksbills, two (B.Dhonfanu, K.BHHR) fit the same model: random emigration, indicating a more transient association with these reefs, no temporal variation in apparent survival, and intra-population variation in detection probability (S2 Table). L.Hithadhoo fit a model with no temporary emigration between six month periods (indicating a strong association with this reef), intra-population variation in detection probability, and no temporal variation in apparent survival. A.Dhidhdhoo fit a similar model, but one that

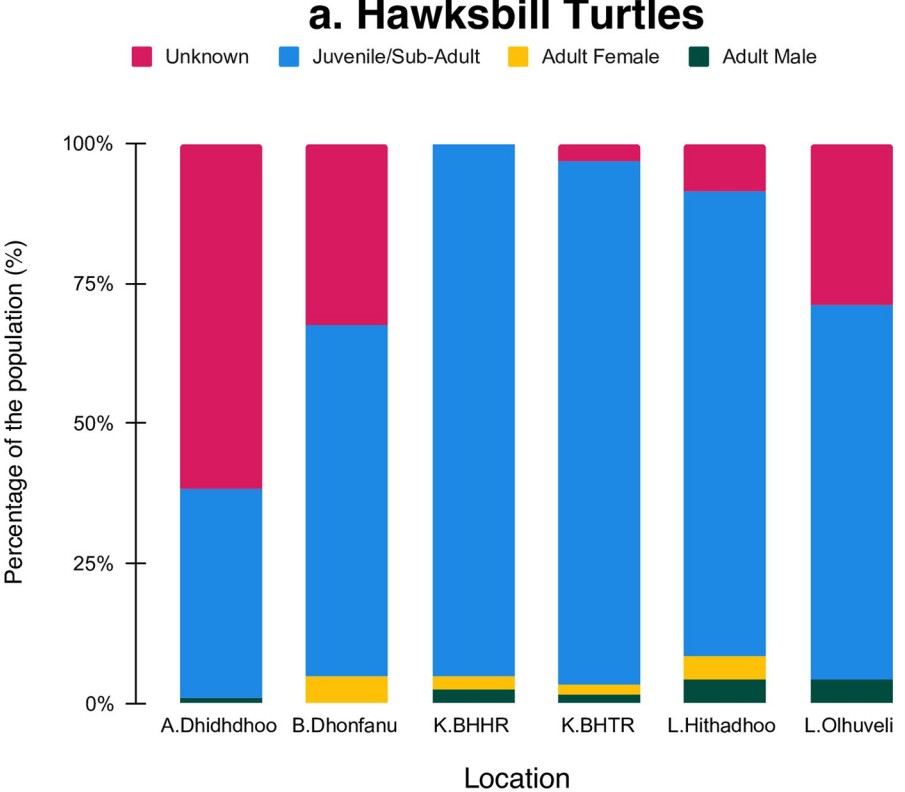

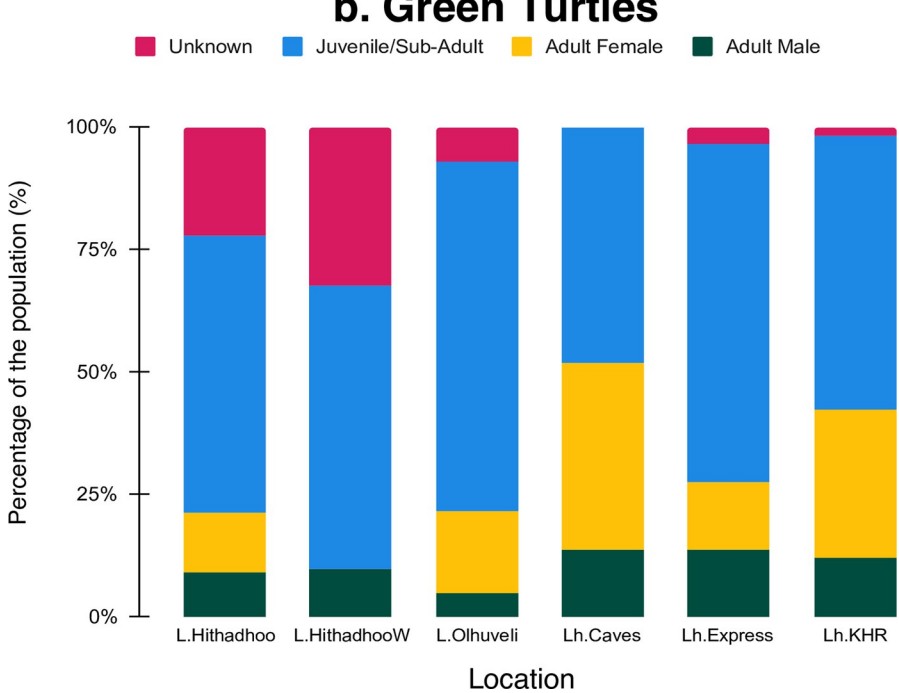

**Fig 1.** Sex and age class of a. hawksbill and b. green turtles at the 10 reefs analyzed for this study, see S1 Table for site code definitions.

involved monthly and seasonal variation in detection probability across all turtles. L.Olhuveli also fit a similar model to L.Hithadhoo, but with homogenous detection probability across all turtles. Lastly, K.BHTR fit a model of Markovian emigration (meaning that turtles associated with these sites are less likely to leave and turtles not associated with these sites are less likely to become associated with them at the next time step, see [44, 45], intra-population variation in detection probability, and no temporal variation in apparent survival. Two of six models failed at least one test for overdispersion (A.Dhidhdhoo, K.BHTR), the former for heterogeneous detection across turtles, and the latter for both a trap effect and heterogeneous apparent survival rates.

Out of the six reefs examined for greens, four (Lh.Express, L.Hithadhoo, L.HithadhooW, and Lh.KHR) fit the same model based on the lowest (Q)AIC, one with no temporary emigration between six month periods (indicating a strong association with home reefs), intra-population variation in detection probability, and no temporal variation in apparent survival. Lh. Caves fit a similar model, but one that involved homogenous detection probability across all turtles. L.Olhuveli fit a model with random emigration, indicating less consistent reef association, seasonal variation in detection probability, and monthly and seasonal variation in detection probability (S2 Table). Three of the six models failed at least one test for overdispersion (Lh.Express, L.HithadhooW, and L.Hithadhoo), where the first two displayed heterogeneous detection rates and the last displayed heterogeneous detection and apparent survival rates.

For hawksbills, the estimated population increased at three of the six reefs (A.Dhidhdhoo, L.Hithadhoo, and L.Olhuveli), with the greatest rise at A.Dhidhdhoo, which more than doubled its estimated population during the study period (Fig 2A). For greens, the estimated population increased at all six reefs with the greatest rise of over an order of magnitude at Lh. Hithadhoo during the study period (Fig 2B).

Hawksbills exhibited fluctuations in estimated population between the monsoon seasons for most reefs (Fig 2A). These fluctuations in estimated population were particularly prevalent at L.Hithadhoo, L.Olhuveli, and K.BHHR, but were also observed to a lesser degree at K. BHTR. A pattern of increasing estimated population in the dry season (November to April) and decreasing estimated population in the wet season (May to October) repeated in 2016, 2017, 2018, and 2019, with the exception of 2018 at K.BHHR, where the population remained constant over a 12-month period. The opposite seasonal pattern was observed at B.Dhonfanu. A different pattern was observed at A.Dhidhdhoo, where estimated population increased over a 12-month period, followed by a decrease in estimated population over the subsequent 12 months. In contrast, there were no obvious seasonal fluctuations in estimated population that repeated throughout the entire time series for green turtles, except at L.Olhuveli where there was a pattern of decreasing estimated population in the dry season and increasing estimated population in the wet season (Fig 2B). At L.Hithadhoo and L.HithadhooW, there was a pattern of decreasing estimated population in the dry season and increasing estimated population in the wet season that repeated in 2016 and 2017, but reversed in 2018, and did not repeat in 2019. At Lh.Caves and Lh.Express, a pattern of increasing estimated population in the dry season and decreasing estimated population in the wet season was observed until May 2018, after which no seasonal fluctuation in estimated population could be discerned. No seasonal fluctuation in estimated population was detected for any year at Lh.KHR.

For hawksbills, apparent annual survival ranged from a low of 55.8% (95% confidence interval 40.8–69.1%) at L.Olhuveli to a high of 82.0% (95% confidence interval 66.5–91.0%) at K.BHTR (Fig 3A). The mean annual apparent survival rate of hawksbills across all six reefs was 68.6% (95% confidence interval 45.1–68.0%). Annual apparent survival for green turtles ranged from a low 54.0% (95% confidence interval 40.0–66.9%) at L.Olhuveli to a high of 99.8% (95% confidence interval 66.3–83.7%) at L.Hithadhoo (Fig 3B). Mean annual apparent survival for green turtles at all six reefs was 80.3% (95% confidence interval 50.9–87.4%).

## a. Hawksbill Turtles

## b. Green Turtles

**Fig 2.** Estimated population curves for a. hawksbill and b. green turtles between May 2016 and November 2019. The first time interval was not included for Lh.Express due to paucity of data. See S1 Table for a list of site codes.

## a. Hawksbill Turtles

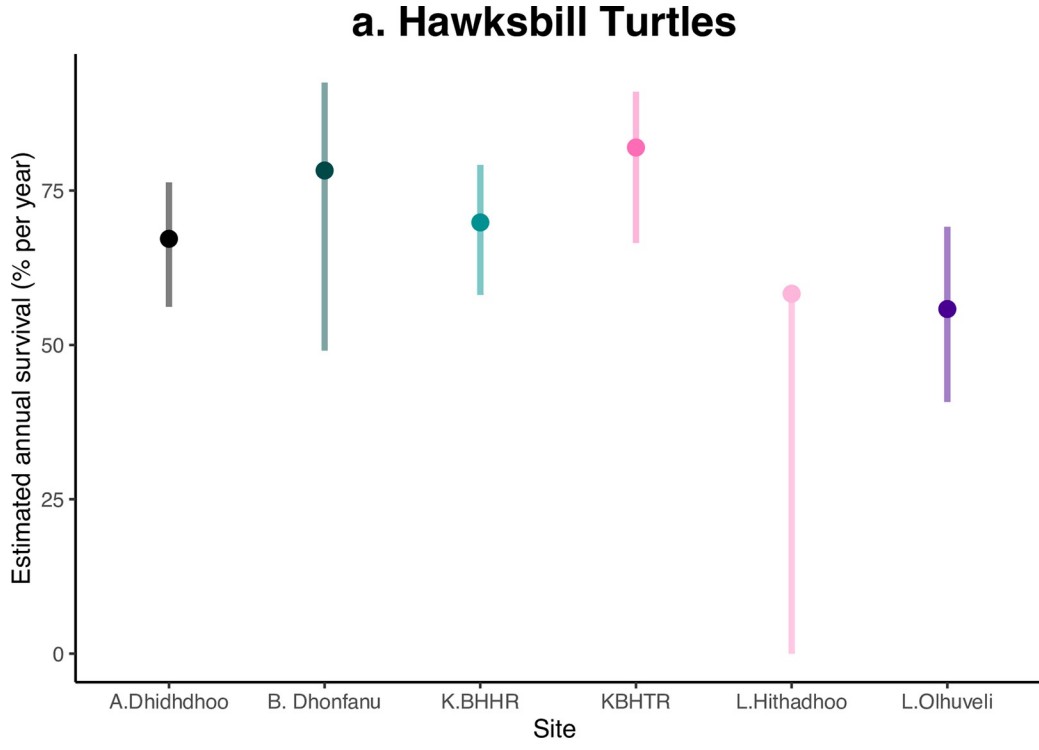

## b. Green Turtles

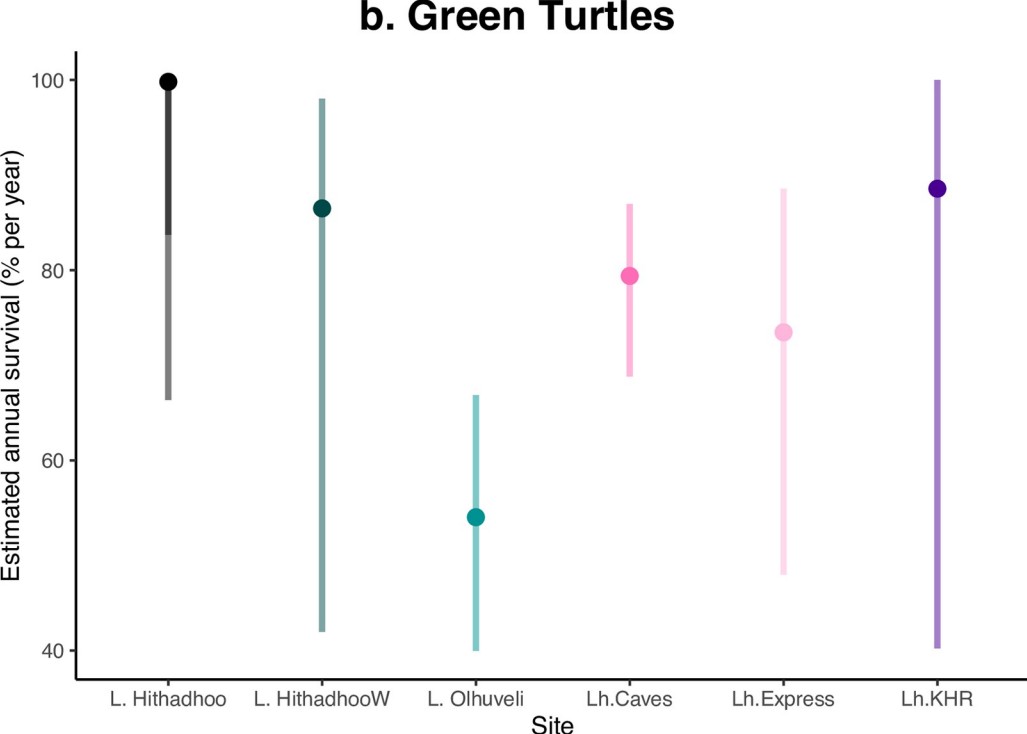

**Fig 3.** Annual apparent survival for a. hawksbill and b. green turtles at six reefs, see S1 Table for site code definitions.

For hawksbills, L.Hithadhoo had the largest increase in estimated population but a relatively low coefficient of variation (CV), indicating an increasing but stable population at this site (Fig 4A). A.Dhidhdhoo and L.Olhuveli had moderate increases in abundance but higher CVs, indicating a population that is slowly increasing but more unstable. B.Dhonfanu, K. BHHR, and K.BHTR all showed multiyear (3.5 years) decreases in abundance below 0%. K. BHHR and K.BHTR had lower CVs, indicating shrinking but stable populations at these sites, while B.Dhonfanu had a relatively high CV, indicating that its population is shrinking but less stable. For green turtles, Lh.KHR had the highest multi-year increase in abundance but a moderate CV, indicating relative instability in the population (Fig 4B). L.Hithadhoo had a moderate increase in abundance, with the lowest CV, indicating the most stable green turtle population. L.Olhuveli had the lowest multi year increase in abundance and the highest CV, indicating a population that is slowly increasing in abundance but one that is very unstable. Compared to hawksbill turtles, green turtles had higher CVs, indicating that their populations are more unstable. L.HithadhooW, Lh.Caves, and Lh.Express had low to intermediate multi year increases in abundance and low to moderate population stabilities.

## Discussion

Our linkage of opportunistic data with capture-mark-recapture models has allowed us to uncover key trends in Maldivian sea turtle demography. Juvenile green and hawksbill turtles made up the majority of individuals recorded in this study. Preliminary analyses suggest that the populations of both species are stable and/or increasing in abundance in the short term at many reefs in the Maldives. Annual apparent survival was higher for greens, but comparatively, hawksbills showed greater stability in their populations.

Previous studies suggest temperature as well as benthic structure as important factors governing habitat use [51–53], which might explain the differences in abundance we recorded across atolls. Hawksbills in the Arabian Gulf have been documented to migrate to deeper water during hotter months [53]. This would coincide with the fluctuation observed on B. Dhonfanu, which is an island on the inside of an atoll. In contrast, Hawksbill populations on L.Hithadhoo and L.Olhuveli have been observed to follow the opposite pattern. This might be associated with greater reef depth (particularly at L.Hithadhoo), as well as the location of both sites near a channel, which may provide an influx of colder water. Benthic structure might play an additional role in combination with shifting currents between monsoon seasons [52].

Previous CMR studies using photo-ID as well as traditional tagging methods reported survival rates within the same range as ours for adult loggerheads [54], juvenile loggerheads [55], and juvenile to adult green turtles [56]. Previously estimated survival rates of green turtles in the southern Great Barrier Reef were higher for adults than for sub-adults and juveniles [56]. As the majority of individuals identified in our photo-ID study and assigned to an age class were juveniles, we anticipated apparent survival rates would be more reflective of previously published juvenile survival rates, which holds true in our case across sites and species. In addition to 'apparent mortality' due to emigration, our reported apparent survival rates can be assumed to be conservative (e.g., underestimates) estimates of actual survival, as individuals might still be present but persistently no longer photographed (e.g., due to learned avoidance behaviour), similar to the possibility of tag loss in other CMR studies [55].

The observed lower apparent survival for hawksbill turtles might thus be related to the higher number of juvenile individuals in the known population. In addition, the potentially less stable association of juveniles with a respective habitat [57–59], and/or various environmental or disturbance-related factors might also contribute to the observed difference.

## a. Hawksbill Turtles

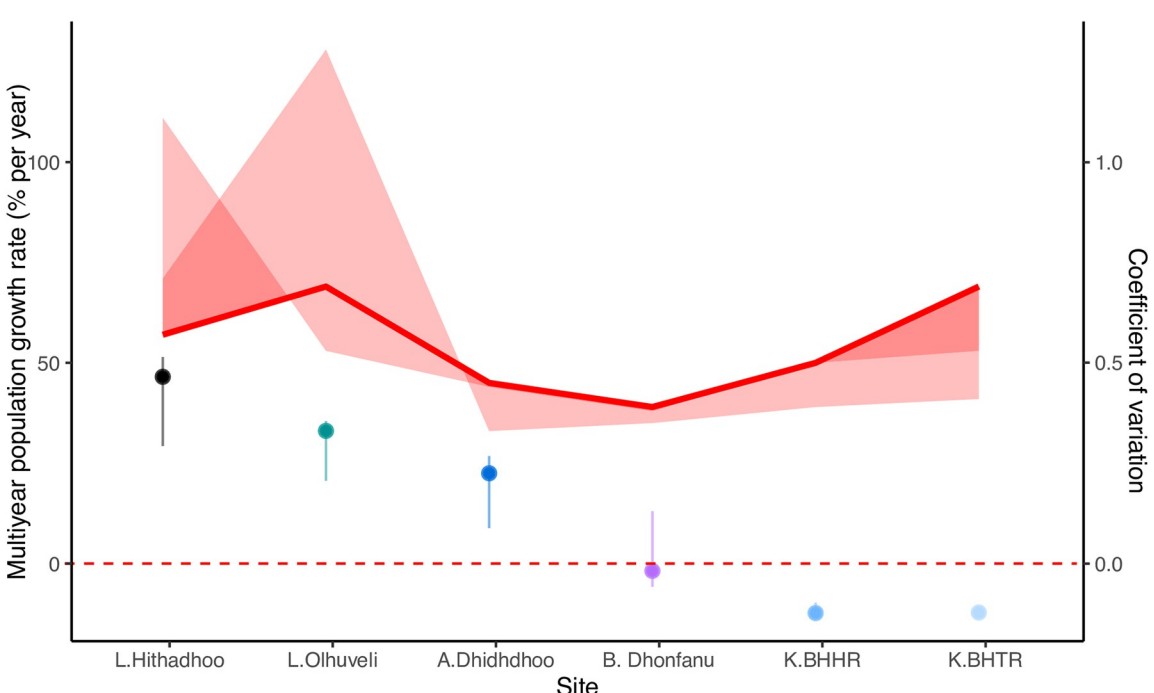

## b. Green Turtles

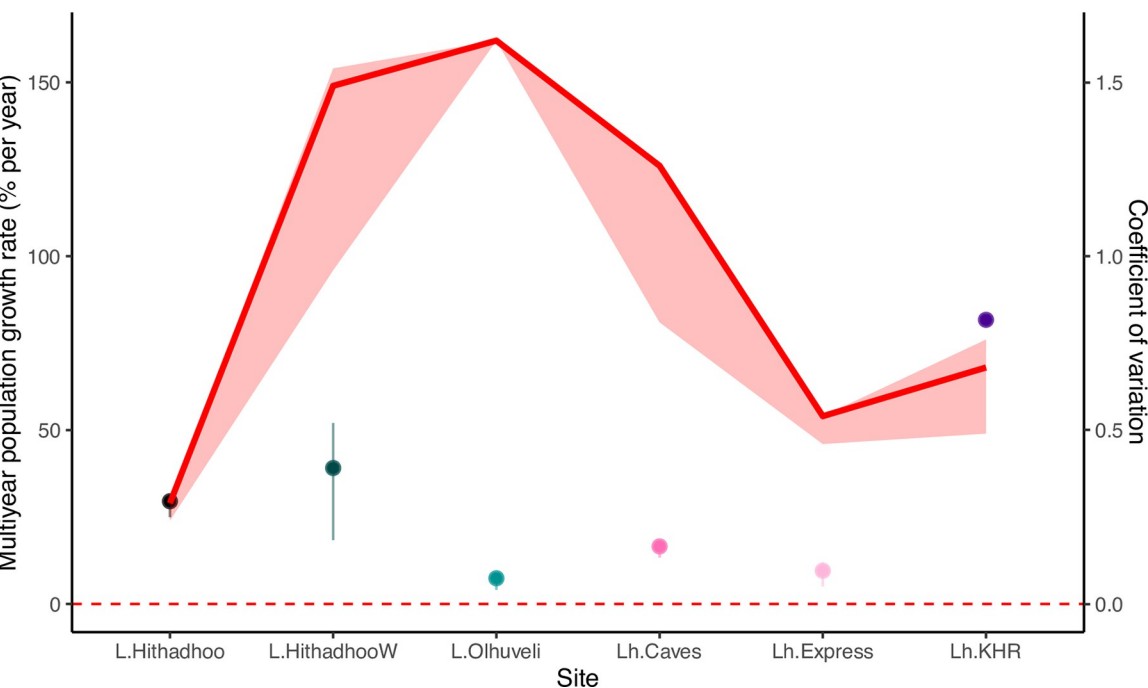

**Fig 4.** The multiyear increase in abundance (over 3.5 years in most cases) and the coefficient of variation (CV) (a representation of the stability of the population) for a. hawksbill and b. green turtles across the 10 reefs. The dashed red line differentiates between an increasing and decreasing population (0% change in abundance). See S1 Table for site code definitions. Mean coefficient of variation is shown as a solid red line, and the 95% confidence interval is shown as a shaded red area.

In this study, we did not photograph any turtles in more than one atoll and only in rare cases (<1% of the population) were turtles photographed on multiple reefs more than 2 km apart. Our findings agree with previous reports of high site fidelity in both greens and hawksbills [17, 27–31, 60]. These results must be interpreted with caution given our restricted set of examined sites. Moreover, a future increase in survey effort might reveal currently unknown intra- and inter-atoll migration, which might contribute to some of the emigration observed in our study.

We found a range of emigration dynamics across sites. Most sites exhibited a lack of emigration, which indicates high capacity for sites to support new immigrants, as there is no need for turtles to leave sites (e.g., due to nutrient limitation). At the sites where random emigration was observed, it may indicate decreased affinity or preference for the given sites. In contrast, the hawksbill population at K.BHTR exhibited Markovian emigration, which indicates two things: 1) that turtles are more likely to still be at the reef at the next time period if they were present at the current time period, and 2) that turtles absent from a reef at a given time period are likely to stay absent at the next time period [45]. This may be due to local adaptation to site-specific characteristics, or territoriality of individuals at this site, but further examination is required to better understand the mechanism for these dynamics.

We would expect adult turtles foraging in the Maldives to leave the area at certain intervals for breeding migration, which would be reflected as random emigration in our models. Currently, we cannot capture this temporary migration at any of the reefs studied. This is most likely because adults only comprise a small portion of our dataset and the site fidelity and/or emigration of the larger group of immature turtles is dominant. It is currently unclear why sea turtles are abandoning seemingly viable foraging habitat, and switching to different foraging areas, without clearly identifiable triggers. As a large portion of the identified turtles were juveniles, the Maldives might be serving as a recruitment area for both species, similar to other island habitats in the Indian Ocean [61],and providing developmental habitat for immature turtles. Previous studies have documented the phenomenon of immature turtles switching foraging habitats, and have discussed both individual preferences of turtles as well as environmental conditions as influences on this behavior, but no clear explanation has been found so far [31, 57–60]. The exact reasons for differences in site fidelity remain unclear (see for example [60]).

The Maldives appears to provide excellent habitat for recruiting juvenile hawksbill and green turtles. Upward trends in abundance found in this study corroborate similar studies on sea turtle populations in the region; for example, greens and loggerheads in the South-West Indian Ocean, olive ridleys in the North-East Indian Ocean [62] and greens and hawksbills in the Chagos Archipelago [63]. In contrast, currently available Red List Assessments for green turtles in the Northern Indian Ocean [8, 64] as well as for hawksbills both globally and within the Maldives [7, 9] found decreasing population trends of nesting populations. Indeed, five of the eleven most endangered turtle regional management units are located in the Indian Ocean [20]. To explain this, we note that IUCN assessments were based on the adult population, whereas our study encompasses all life stages of the Maldivian turtle population, with a particularly high percentage of immature individuals. We expect that different life stages will exhibit different trends, as, for example, adults are more likely to be targeted for poaching, and there is an increased impact of fishing activity in more offshore areas (e.g., during breeding migrations of adults). In contrast, juveniles likely benefit from successful nesting beach protection, which could explain the positive trends seen here. Future integration of life-history-stage-level demographic parameters into population viability analyses could help understand key conservation interventions to benefit population-level trends in the Indian Ocean [65].

Due to a lack of genetic and satellite telemetry data, the nesting population to which sea turtles observed foraging in the Maldives belong remains unknown. Two green turtles nesting in the Chagos Archipelago equipped with satellite transmitters were recorded traveling to the Maldives [66]. This is the only existing study examining the potential genetic link between Maldivian green turtles and a rookery in the Indian Ocean. It remains to be investigated whether the green turtles nesting in the Maldives also reside in the archipelago during non-nesting periods, as no turtle that has been photographed while nesting has also been positively identified foraging and vice versa. An unsuccessful mating attempt by an identified male green turtle considered a Lhaviyani atoll resident with a similarly identified resident female was recorded once; however, no successful mating by either an identified resident male or female turtle has been observed thereafter. Hawksbills remain completely elusive, as no studies into satellite telemetry or genetics for individuals from the Maldives have been conducted. It is highly likely that the Maldivian hawksbill population is connected to other nesting populations in the Indian Ocean basin, since previous genetic and tagging studies have found evidence of genetic homogeneity and long-distance migration for populations as far away from each other as Seychelles and the Cocos Keeling Islands (see [61] for a summary). Combining photo ID data from different parts of the Indian Ocean, as well as the application of other methods such as population genetics, could address the question in the future.

## Study limitations

**Initial data collection.**   The majority of thisstudy's data were provided by various contributors and citizen scientists without a measure of effort included, as we did not receive negative reports. Because life stage identification was based on data submitter-provided size estimates, estimates may vary among observers and have not yet been verified with techniques tested in other marine species, such as paired laser photogrammetry or stereo-image measurements [18, 67]. These techniques should be applied in the future whenever possible to confirm size estimates for consistently monitored subpopulations. While previous reports have shown that underwater estimates of turtle sizes can be accurate or at least within 10 cm of actual turtle size [18, 68], the comparatively small divergence can result in significant differences in the identification of adult female turtles, and estimates might therefore not be reflective of the true sex ratio in the population.

Along these lines, our estimated sex ratios based on size estimates indicate a female biased adult population, which might be the result of a) observers overestimating the size of turtles and falsely identifying immature males as females, b) male turtles using deeper and therefore unsurveyed sections of the reefs, as anecdotal reports on male turtle resting spots indicate, or c) a combination of both effects. Studies of hatchling sex ratios of sea turtle populations around the globe paint a mixed picture of severe female bias in some rookeries [69, 70] and a comparatively balanced sex ratio in others [71].

While the Robust Design framework allows for more realism, it still relies on several assumptions. These include that sites have no emigration during closed periods, and that turtles only disperse during open periods, which are simplifications of turtle site affinity. Nonetheless, we chose to model major population fluctuations as being due to seasonal changes, since major ocean current fluctuations are likely a major driver of sea turtle movement patterns [72]. Secondly, while this framework accounts for heterogeneity in detectability, it does so in a fairly simplified way, by estimating two distinct population subgroups with different detectabilities [22]. More complex detectability functions would improve future realism, and would provide more rigorous support for the environmental drivers we propose for turtle site affinity. Thirdly, this model still cannot account for gaps in survey effort, which required us to

limit our examination to the period of 2016–2019, while *ad hoc* surveys were ongoing for some time prior. Fourth, the model cannot distinguish emigration from mortality, which limits the assessment of risk to turtle species. Nonetheless, the suitable habitat for sea turtle species is declining, and emigration can signal a decrease in habitat quality [73]. Finally, an external validation would be a more stringent test of these models. We did not have enough data to use a holdout set, but a test of model prediction ability remains an important future step in improving the reliability of capture-mark-recapture models with opportunistic data.

## Conclusion

Despite the challenges of species-at-risk monitoring in this region and some study limitations, we successfully applied a Robust Design framework to green and hawksbill sea turtle populations from the Maldives. We were able to identify species-specific differences in demographic parameters and provide a more positive picture of juvenile sea turtle trends in the Indian Ocean in contrast to existing estimates of adult populations. Our results represent one of the first empirical estimations of sea turtle population trends that account for detectability. For countries in the Global South—particularly for small island states, evaluating the severity of population declines can be especially challenging due to lack of resources. The merging of detectability models with citizen science and volunteer-collected data has allowed us to begin to overcome these challenges in a cost-effective way. This work underlines the importance of simple, long-term monitoring programs that can contribute reliable information to help assess the conservation status and efficiency of management strategies. We hope that this work can serve as a framework for future monitoring activities.

## Supporting information

**S1 Fig. Map of the study sites.**
(TIF)

**S2 Fig.** Population estimates with standard error shown at each primary interval for a. hawksbill b. green turtles. See S1 Table for a description of site codes.
(TIF)

**S1 Table. Sites codes and corresponding atoll and centroid coordinates for study sites.**
(XLSX)

**S2 Table. Best capture-recapture population model fit per site and species, based on the suite of models tested in [22].** S(.) = no variation in apparent survival; S(season) = seasonal variation in apparent survival; p(season.month) = seasonal and monthly variation in capture probability; p(session) = monthly variation in capture probability; p(mixture) = individual heterogeneity in capture probability, p(.) = no variation in capture probability. See S3 Table for more information on the suite of models tested and S1 Table for site code information.
(XLSX)

**S3 Table. Robust design models applied to the hawksbill and green turtle dataset, identical to those tested in 22.** The number of parameters (npar) fit for each model is shown. S(season) = seasonal variation in apparent survival; S(.) = no variation in apparent survival; p(season. month) = seasonal and monthly variation in capture probability; p(month) = monthly variation in capture probability, p(mixture) = individual heterogeneity in capture probability, p(.) = no variation in capture probability.
(XLSX)

**S1 Appendix. Details of the models tested.**
(DOCX)

## Acknowledgments

The authors would like to thank all data submitters and citizen scientists who contributed photos to this project as well as cooperation partners in Lhaviyani and Laamu atolls, particularly any members of the local Maldivian community.

## Author Contributions

**Conceptualization:** Jillian A. Hudgins.

**Data curation:** Jillian A. Hudgins, Stephanie Köhnk.

**Formal analysis:** Jillian A. Hudgins, Emma J. Hudgins.

**Investigation:** Jillian A. Hudgins, Stephanie Köhnk.

**Methodology:** Jillian A. Hudgins, Emma J. Hudgins.

**Project administration:** Jillian A. Hudgins, Martin R. Stelfox.

**Software:** Emma J. Hudgins.

**Supervision:** Martin R. Stelfox.

**Visualization:** Emma J. Hudgins.

**Writing – original draft:** Jillian A. Hudgins, Emma J. Hudgins, Stephanie Köhnk, Enas Mohamed Riyad, Martin R. Stelfox.

**Writing – review & editing:** Jillian A. Hudgins, Emma J. Hudgins, Stephanie Köhnk, Enas Mohamed Riyad, Martin R. Stelfox.

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
