## [Decision Letter · Decision Letter 0]

2 Aug 2022

PONE-D-22-19155A brighter future? Stable and growing sea turtle populations in the Republic of MaldivesPLOS ONE

Dear Dr. Hudgins, Thank you for submitting your manuscript to PLOS ONE. After careful consideration, we feel that it has merit but does not fully meet PLOS ONE’s publication criteria as it currently stands. Therefore, we invite you to submit a revised version of the manuscript that addresses the points raised during the review process. Please follow the referee suggestions for how you need to rewrite this manuscript so that the writing is not so parochial and has broader appeal.

We look forward to receiving your revised manuscript.

Kind regards,

Graeme Hays

Academic Editor

PLOS ONE

Journal Requirements:

Additional Editor Comments:

We two detailed and constructive reviews of your manuscript. Both referees recommend that major revisions are needed. If you take care with a revision, this manuscript might still be suitable for PLoS1 and so I am recommending that you revise the manuscript taking all the comments into consideration. I expect that this major revision will take some time as a lot of work is called for.

I look forward to seeing a revision.

All best wishes, Graeme Hays

Reviewers' comments:

Reviewer's Responses to Questions

**Comments to the Author**

1. Is the manuscript technically sound, and do the data support the conclusions?

Reviewer #1: Yes

Reviewer #2: Partly

2. Has the statistical analysis been performed appropriately and rigorously? 

Reviewer #1: Yes

Reviewer #2: No

3. Have the authors made all data underlying the findings in their manuscript fully available?

Reviewer #1: Yes

Reviewer #2: No

4. Is the manuscript presented in an intelligible fashion and written in standard English?

Reviewer #1: Yes

Reviewer #2: Yes

5. Review Comments to the Author

Reviewer #1: Here photo id is used to estimate numbers of hawksbill and green turtles at foraging sites in the Maldives. Using mark-recapture approaches, estimates of survival and emigration are made. It is concluded that at many of the sites the numbers of turtles are stable or increasing.

This manuscript contains some interesting observations and nice analysis. The authors should be applauded for how they have collected an extensive photo-id database. While the data collection and analysis are nice, it is a shame that the manuscript is let down badly by the writing. At the moment the Discussion is poor, being almost entirely very parochial writing or repetition of the Results. So almost the entire Discussion need to be rewritten. I appreciate the authors may be new to academic writing, so I have tried to provide helpful comments to guide them about how to put together a Discussion. So a major revision is needed and the authors need to take great care placing their work into a much broader context. If the authors can make these improvements, then their work will be far more widely read and have much greater impact. Throughout the authors need to identify their key results and the discuss those results in the context of the broader literature. E.g. “Our findings that X,Y,Z occurred is similar/contrasts to what has been found in other studies around the world. For example in study A it was found X, while we showed Y …”

Throughout it is frustrating that the authors have not included line numbers or even page numbers. I am surprised the manuscript was allowed to go to review in this state.

1. Abstract. “… mean apparent survival estimate of 82.5%±2.6% …”

There needs to be an element of TIME in a survival estimate. You should make your values estimates of annual survival so you can place in the context of wider values. See for example, Fig 2a in: “Schofield G et al. (2020). Long-term photo-id and satellite tracking reveal sex-biased survival linked to movements in an endangered species. Ecology 101, e03027. https://doi.org/10.1002/ecy.3027”

2. Abstract. “… concurrent with recent studies on nesting.”

Delete. Nothing is presented on nesting.

3. Introduction. “The lack of information on adult males is now worrying, due to the concerns of hatchling sex ratios that are becoming more heavily female-biased due to increasing temperatures at nesting beaches worldwide (24; 25).”

You set this up in the Introduction but never return to this topic in your Discussion. What were your adult sex ratios for each species ? Discuss your findings. There are reasons males might not be as rare as previously thought – see “A review of how the biology of male sea turtles may help mitigate female‑biased hatchling sex ratio skews in a warming climate. Marine Biology 169: 89. https://doi.org/10.1007/s00227-022-04074-3”. Did you find balanced adult sex ratios ? Elsewhere in the region, fairly balanced hatchling sex ratios have been estimated due to shading of nests and rainfall. See: “Male hatchling production in sea turtles from one of the world’s largest marine protected areas, the Chagos Archipelago. Scientific Reports, 6:20339. http://dx.doi.org/10.1038/srep20339.”. So skewed sex ratio might not be such a big problem in the Indian ocean. Discuss your findings in this context.

4. Introduction. Fidelity.

“Many previous studies have revealed the high site fidelity of both species (29; 30; 21; 31; 32).”

For a review of green turtles, see: “Shimada (2022). Fidelity to foraging sites after long migrations. Journal of Animal Ecology 89, 1008–1016. https://doi.org/10.1111/1365-2656.13157.”. Here you need to DISCUSS your findings in the context of the broader literature. What do your findings add ? When your turtles leave, do they subsequently return or do they leave for distant areas – see both of these scenarios in long-term tracking of hawksbills in the region: “High accuracy tracking reveals how small conservation areas can protect marine megafauna. Ecological Applications, 31(7), e02418. https://doi.org/10.1002/eap.2418”.

Or are they dying and so being removed from the population ? All this needs to be discussed in the context of the broader literature.

5. Table 1. Not needed in main text. Put in SI.

6. Life stages. Methods How were adult identified ? e.g. how did you distinguish a large sub-adult from an adult ? So presumably you estimated SIZE of turtles. Detail how this was all done. Be honest – if you are not really sure of adults V large sub-adults then say this. Turtles have been sized by divers previously in the region: “Houghton et al. (2003). Habitat utilisation of juvenile hawksbill turtles (Eretmochelys imbricata) in a shallow water coral reef habitat. Journal of Natural History 37, 1269-1280.”. Discuss your approach (in the Discussion), i.e. how good are your life-stage and size estimates ?

The following includes results you then need to discuss (in the discussion) in the context of the broader literature:

7. Life stages. What was the adult sex ratio ? Place in the context of the broader literature.

8. Where do the adults come from ? Place in the broader context. For example, satellite tracking has shown that some nesting green turtles from the Chagos Archipelago migrate to foraging grounds in the Maldives. See Figure 1 in: “Open ocean reorientation and challenges of island finding by sea turtles during long-distance migration. Current Biology 30, 3236–3242. https://doi.org/10.1016/j.cub.2020.05.086”. So some of the connections are known. Discuss. In contrast no tracked nesting hawksbill turtles from the Chagos Archipelago have travelled to the Maldives (see: “Travel routes to remote ocean targets reveal the map sense resolution for a marine migrant. Journal of the Royal Society Interface 20210859. https://doi.org/10.1098/rsif.2021.0859”), suggesting the adult hawksbills you see might be more locally derived ? Discuss.

9. Inferred emigration. Place in the broader context. For example long term tracking has shown protracted residence of immature hawksbill turtles in localised areas in the Chagos Archipelago while only a few individuals relocate. See: “High accuracy tracking reveals how small conservation areas can protect marine megafauna. Ecological Applications, 31(7), e02418. https://doi.org/10.1002/eap.2418”. See references within that paper for the broader literature from other sites. Place your observations into that wider context.

10. “For hawksbills, apparent survival ranged from a low of 76% at …”

Place these findings into the broader context. For example, survival rates have been measured by photo id and other mark recapture approaches for a range of species. E.g. as a starting point see: “Long-term photo-id and satellite tracking reveal sex-biased survival linked to movements in an endangered species. Ecology 101, e03027. https://doi.org/10.1002/ecy.3027”. Discuss your findings within this broader context.

11. “… the highest growth rates but …”

This reads as you were recording the growth rate of individuals. Rephrase. Here I think you are talking about an increase in abundance. Rephrase throughout rather than talking about “growth”.

12. Figure. “Estimated survival (%)” This is a strange unit. Survival needs to be calculated in % surviving per year (annual survival rate) and placed into a broader context. So extrapolate you measurement interval up to 365 days to get annual survival values.

13. What are trends in numbers of these species elsewhere in the region. E.g. as a starting point see Mortimer et al. Estimates of sea turtle nesting populations in the south-western Indian Ocean indicate the importance of the Chagos Archipelago. Oryx 54, 332–343. https://doi.org/10.1017/S0030605319001108. E.g. are encouraging upward or stable trends seen elsewhere in the Indian Ocean for hawksbills and greens. Place your results in this broader context. Are your finding consistent or different to what other have recently reported ?

14.. Discussion. “Hawksbills in the Arabian Gulf have been documented to migrate …”

This discussion needs to be expanded. What have others reported ? e.g. “Ecological Applications, 31(7), e02418. https://doi.org/10.1002/eap.2418”.

The following are long sections of the Discussion that can be deleted as they are just background information or repetition. Instead build you discussion around discussing your results in the context of the broader literature.

15. Discussion. Repetition of Results. Delete: “We also hypothesized that estimated hawksbill abundance and stability would be higher than green turtle abundance and stability. This held largely true, indicating a greater relative need to protect green turtles, though we note that green turtle populations are estimated to be growing at all sites, while several sites had shrinking hawksbill populations.”

16. Discussion. Repetition of Results. Delete:

“The hawksbill population at L.Hithadhoo had a high multiyear growth rate and relatively low CV, indicating it was home to a fast growing, stable population of hawksbill turtles. This site is best described by a model with no temporary emigration, indicating that turtles arrive at this reef and remain there. The populations at A.Dhidhdhoo and L.Olhuveli were also growing relatively quickly but their CVs were higher, indicating more instability in the populations at these sites. Both L.Olhuveli and A.Dhidhdhoo fit a model with no temporary emigration, though modeling for A.Dhidhdhoo showed temporal variation in detection probability across all turtles. Populations at K.BHTR and K.BHHR showed negative growth rates but relatively low CVs, indicating shrinking but relatively stable populations at these reefs. B.Dhonfanu also showed a negative growth rate along with a high CV, indicating an unstable, shrinking population. In addition, our results indicated a transient association with this reef. This could indicate that the turtles were leaving the reef due to disturbance.”

17. Discussion. Very parochial and repetitive of Results. Delete: “The Dreaming of Maldives resort repository (dreamingofmaldives.com) indicates that several new resorts have been built in the Maldives in the last few years, particularly within Baa atoll, which would have caused noise and sedimentation during construction, as well as increased boat traffic and tourist presence. These additional stressors may have caused the unstable and transient nature of the turtle population at this site. Out of the six reefs analyzed here, one reef had a fast-growing stable population, two had stable shrinking populations, two had growing but unstable populations, and one reef had an unstable shrinking population. The overall development of the population of the country is still not entirely resolved. The mixed results from the currently investigated reefs could indicate a shift in habitat use, with turtles moving from sites with higher disturbances to other reefs. Currently, no such movement has been documented with the photo ID method, but this could be due to the continued under- sampling of reefs in the country.”

18. Discussion. Repetition of Results. Delete. “Green Sea Turtles

Lh.KHR had the highest growth rate but a relatively high CV, indicating that the population was growing quickly but was relatively unstable. In addition, Lh.KHR fit a model of no emigration, but individual variation in detection probability, potentially indicating lower detectability at this site. There was high recruitment to Lh.KHR, but some turtles appear more challenging to relocate. Lh.KHR features extensive seagrass meadows, which provide an excellent foraging area for green turtles. It is possible that the observed population trends are an indicator for the function of this reef as a recruitment area and first stepping stone for greens coming into the archipelago. In total, the area has a large carrying capacity, but greens occasionally move on after some time. The difference in detection probability potentially resulted from individual variation in which area of the seagrass bed was used by the turtles. The first section of this site the lagoon closest to the jetty and main activity area is frequented more often than the second section further down the beach, resulting in different use by citizen scientists (i.e., tourists). L.HithadhooW and L.Hithadhoo both had relatively high growth rates, indicating that many turtles are immigrating to these sites. Both sites also fit models with no emigration, indicating strong retention of individuals across seasons. However, L.Hithadhoo displayed more variable detection, indicating potentially more sporadic surveying, while the former may display more unstable recruitment dynamics.”

19. Discussion. Repetition of Results. Delete. “L.Olhuveli had a low growth rate but a relatively high CV. It also had the lowest apparent survival of the reefs analyzed, indicating greater instability compared with other reefs, and a potential decline in habitat suitability over time. However, it fit a model with no emigration. There may have been some disturbance at this site, such as seagrass removal at nearby meadows that occurred up until 2018 (55). This could warrant further investigation, or it may indicate that carrying capacity of the reef (saturation) has been reached. Lh.Caves had a relatively low growth rate and higher CV, indicating that this population was moderately unstable but growing slowly. Finally, Lh.Express had the lowest growth rates but one of the lowest CVs, indicating that this population was stable and growing slowly. Modeling results also indicated that turtles were remaining at these reefs with strong fidelity.”

20. Repetition. Delete. “Out of the six reefs analyzed here, one reef had a fast-growing unstable population, one had an unstable slowly growing population, while the remaining four reefs were growing moderately and were moderately stable. However, we note that we only have 3.5 years of data over which we can ascertain trends, and longer-term datasets may reveal different growth and stability patterns.”

21. Background information. Not needed. Delete “Further Research

Determining population health of greens and hawksbill sea turtles without first defining their populations’ boundaries and management unit(s) (MU) comes with its challenges. Whilst our results are promising, we do not know how fragmented the MUs are. Future research should focus on uncovering the genetics of sea turtles in the country to better evaluate the impact of various threats. Moreover, improved phylogeography of green and hawksbill sea turtles in the Maldives will help resolve metapopulation dynamics in the region (56).”

22. “Policy and management implications”

Delete this section. It is all background information and there is no discussion on your results in this context.

23. Conclusion. This long section is not needed. It repeats the Abstract. Reduce to 2 or 3 lines at most.

24. references. Inconsistent reference formatting. Take some care and put references in the correct style.

25. No figure legends in the pdf.

26. Figure 2. Always include some units on figure. So “estimated population size (number of individuals)”

27. Figure 3. Needs to be annual survival rates. Current units are meaningless.

28. Figure 3. Why are a and b needed ? Combine ? Hard to tell without a legend.

29 Figure 4. Growth rate should be “increase in estimate population size”. And units should not be “%” but need some element of time, e.g. % per year.

In summary, with some care and careful rewriting this could be a solid contribution. Graeme Hays

Reviewer #2: General

This is a really exciting paper, with some really useful insights on foraging sea turtles in a region with limited research. However, at present, this manuscript is set out as a report for a local management agency, rather than a broad scientific audience. I have made detailed comments throughout – do, look at my suggestions on the figures with the results combined. There is a lack of information on the robustness of the raw dataset and how issues in seasonal variation in data collection were overcome. The results are too detailed, with it being very difficult to interpret the key findings. The introduction and discussion also need broadening to draw on the existing literature on the general concepts covered in this study. I would very much like to see this work published, however, at present major revision is recommended.

Abstract

This is currently written as a report format, and needs a much clearer focus. First, what is the background reasoning for the study, i.e. is there limited data on foraging of sea turtles, and why is it of use to record recruitment and survival – in general, not specifically your site. What is the overarching aim – as you seem to have many, and then you add in an extra one regarding the tourists etc. There is a paper in itself just on how you standardise the ad hoc data before using it to infer survival etc, as resightings will be biased towards hotspots frequented by divers or tourists etc. You state how many records, but not how many unique turtles these represented of each species, and what range of unique individuals were seem across years - this is essential information that needs to be briefly presented to give context to the data.

There are too many results, and too much site specific information – what are the key trends – i.e. your overarching results should be stating whether you have similar numbers of greens and hawksbills, how do immigration, emigration and survival compare for greens and hawksbills compare – are there similarities/differences overall. The seasonal variation is very interesting, do both species exhibit this? The site specific idiosyncracies should be kept for the results/discussion, unless they have a profound impact on general trends.

The wider relevance of the study is unclear, you state that your study shows that the different sources of info are key, but you do not actually evaluate this or the effect of these different sources and potential bias, so I think this detracts from your findings.

To me the key message is that turtles of both species exhibit seasonal use of your region for foraging – it would be interesting to see if this links to their respective food availabilities.

Introduction

I would advise revising this entire section to something like

Paragraph 1 – why it is important to understand foraging behavior of wildlife – highlight seasonality of this – what drives this – climate, depletion of food resources etc… potentially increasing exposure to threats from human activities etc..

Paragraph 2 – issues in marine environment with monitoring immigration, emigration, survival, esp for sea turtles, with disparate foraging areas, seasonality of use, different behaviours adults/juveniles (many items to list here) – how various studies are overcoming this with different techniques (identify some), leading into photo id and citizen science, including issues

Paragraph 3 – sea turtles and foraging – different species forage on different substrate – draw on recent studies looking at foraging behaviour, patch to region use, how multiple species share foraging habitats (several great studies in Gulf of Mexico coming out)

Paragraph 4 – Study aim, objectives, hypotheses tested, wider anticipated importance

Again, this starts too turtle/region/site specific – I would advise changing the focus to seasonality of foraging by different wildlife, particularly herbivores and omnivores, this then leads into a paragraph on turtles and different foraging habits of different species etc. Many of the items drawn on in the introduction have broader relevance, but at present this is not captured.

Photo id – again, this is used widely for many wildlife, so again this can start far more broadly, with a clear explanation of the benefits. There are several recent studies drawing on citizen science to help build turtle databases, and other marine wildlife, so these should be integrated, along with potential issues in bias and how they are being overcome.

“Little historical data exist on marine turtle populations in the Maldives, which makes it difficult to study trends or evaluate the effectiveness of conservation measures put in place by the government” – be careful with this statement. I would advise deleting it, and instead focusing on what information has been published. At present, this paragraph is not set out as an introductory paragraph for a manuscript – this would fit better as an initial study region paragraph in the first part of the methods.

Methods

“visibility is lower during the wet season and there are fewer tourists in the water.”

This is really important, as both issues can be driving variability in your observations.

“Sea turtles can be individually identified based on their arrangement of facial scales.” – this needs to be supported by source citations from the peer reviewed published literature.

“no harm to, or harassment of, turtles” – how can you confirm this, how close did people get to turtles to photograph facial scutes?

“We found large differences in survey effort around the country, and high turnover of staff at many locations, leading to data gaps and lack of data continuity”

How did you address this issue in your models?

“it was approached slowly and as many photographs as possible were taken of

the animal”

How close?

Photo database – who compiled this? Was it a single person or multiple people? What was the false positive/false negative rate? How were such issues overcome? Did you have some way of validating this process?

“Incomplete data (missing identification, dates, or sites) were excluded from analysis” – more details are needed here; how much data did you start with, how many were removed? What % was retained?

In this study, six months (May to October and November to April) was – do you mean two six-month periods? This is not a six month period only, from what I can see.

For the model, first you need to clarify how bias and gaps were addressed/overcome.

Table 1 – this is not a main table, place it as online supplement.

Survival – you have a very short study period – can survival really be detected from this? Especially as your turtle numbers are very low at most sites.

Results

This is methods info:

We restricted site-level analyses to four atolls given the lower number of total turtles at the remaining

atolls. Ten reefs across four atolls were chosen for analysis

Sentence beginning “In total, between 2016 and 2019 to end of paragraph” – the information the reader needs here is

1. How many records for each species and how many individuals overall

2. How were these split across the years – give mean and sd and range, so the reader can perceive the spread

3. How these were split by season – mean and sd and range for the two 6 month periods

4. How these were split by atoll – mean and sd overall, so the reader can perceive the spread – generate a table of the list you gave and refer to this for the individual atolls – these will have limited meaning to a general readership

Life stages

This section is too wordy and too site specific. You have 6 atolls, I just need to know if there was an even spread in life stages or not for the two species. Present the detailed information as a graph and/or supplementary table. Capture the broad trends. If you want to keep site specific details, have them as online supplementary material.

Model interpretation, I would advise simply naming them reef 1 to 6, and clarifying which is which in supp table 1. This interpretation is very complicated. Again, it needs simplifying – what is the key overarching finding, what pattern did most reefs exhibit for each species compared to each other. There is some really interesting information, but it is lost in the detailed information. Place the details as online supplement and draw out the key points of interest for immigration/emigration and then survival – so there should be two clear simple sections.

Table 2 – make this an online supplement

Discussion – this needs to be revised. While the wider literature is drawn on, there is too much site specific information that detracts from the general broad findings of interest. This should not be separated into hawksbills and green turtles, rather paragraphs exploring emigration/immigration and survival trends as a whole, as well as overlap/segratation in foraging habitat use related to life stage, and the issues with photo id and citizen science encountered, including limitaitons..

A paragraph on survival trends is needed and how your trends fit with the wider published literature – there are many studies that have explored this for adults and immature turtles, you need to build on this.

I would advise 5 key paragraphs, i.e.

Para 1 – key overarching findings (these draw your key results together, and set the scene for the discursive content – comparing your findings to the wider literature in subsequent paragraphs)

Para 2 – distribution across atolls – similarity/difference across species, sex specific differences – overlap – segregation – how does this compare to other regions with mixed species – how does different foraging patterns affect this, how even was distribution

Para 3 – seasonality of habitat use – immigration/emigration how this compares to other locations/turtle species – what is driving it – monsoon, other factors, what drives it at other sites

Para 4 – effect of environment versus sampling effort – potential issues etc. Study limitations

Optional para 5 – management implications – keep it broad, relevant to wider audience (site specific stuff can go in an online supplement)

Para 5 conclusions

Figure 1 – please change proportion to percentage. ON each bar, provide the n value, so it can be objectively compared.

What stands out to me here is that juveniles dominated for both, but that there were far more adult green turtles than hawksbills – indicating food availability is not sufficient for adult hawksbills

Figure 2 – What stands out here is one site is key for hawksbills and three for greens (with one key site for greens overlapping for hawksbills) – so, again ,what is of interest here is what is different at the key sites to the other sites for the two species – bathymetry, substrate, exposure? This is worth exploring.

Is the variation detected to do with variation in tourism/diving activity? Or is it truly a seasonal variation. If the latter, then greens exhibit more variability, which might be attributed to depletion of resources.

Figure 3 – I would be very wary of this graph, as your number of years of surveys is limited, your turtle numbers limited, and sampling effort inconsistent. I would advise acknowledging this in your results section – stating that survival ranged from x to x, but that sample sizes were low. So, when only focusing on the 1 and 3 key habitats of hawksbills and green turtles respectively. So change this to a single graph – it will be far more informative, and useful.

In the main text, it would be useful to state in this section what % of turtles were sighted every year (and once every 2 years, once every 3 years) – this then gives the reader an understanding of the robustness of your 5 year time series. i.e., if 90% of turtles are sighted every year, then fine. But, if 50% are sighted every year, and most return with 2-3 or 4 year intervals, then your 5 yr study timeframe is too short.

Also, how many individuals were recorded at more than one atoll within and across years?

Figure missing – please give a map of the region and where the atolls are in relation to each other; this should also be stated in the main text, what is the minimum and maximum difference. Some atolls might indicate lower survival simply because resources were not present. If turtles move around a lot, then actually your data do not reflect survival but potentially use of many sites, with not all sites necessarily being viable in all years. On this figure, you should also identify other sites where turtles were detected foraging but were excluded to demonstrate the mosaic.

Figure 4 – to me, 5 years is not long term, I am not sure this is viable; it depends on what the likelihood of resighting the same turtles is across years. Plus, this is simply driven by the 1 and 3 main sites for each species.

6. PLOS authors have the option to publish the peer review history of their article (what does this mean?). If published, this will include your full peer review and any attached files.

Reviewer #1: No

Reviewer #2: No

---

## [Author Response · Author response to Decision Letter 0]

2 Feb 2023

A response to reviewers comments was included as part of the upload package.

---

## [Decision Letter · Decision Letter 1]

14 Feb 2023

PONE-D-22-19155R1A brighter future? Stable and growing sea turtle populations in the Republic of MaldivesPLOS ONE

Dear Dr. Hudgins,

Thank you for submitting your manuscript to PLOS ONE. After careful consideration, we feel that it has merit but does not fully meet PLOS ONE’s publication criteria as it currently stands. Therefore, we invite you to submit a revised version of the manuscript that addresses the points raised during the review process.

We look forward to receiving your revised manuscript.

Kind regards,

Graeme Hays

Academic Editor

PLOS ONE

Journal Requirements:

Additional Editor Comments:

Your revised manuscript has now been evaluated by two of the original referees. You will see that the referees appreciate the improvements that have been made and they are generally happy with the revision. Nice work ! One referee suggests a few more minor changes. With a little care, I think you should be able to address all these remaining points and then this will make a very nice contribution to PLoS1.

Thanks again for all your hard work on the revision.

All best wishes, Graeme

Reviewers' comments:

Reviewer's Responses to Questions

**Comments to the Author**

1. If the authors have adequately addressed your comments raised in a previous round of review and you feel that this manuscript is now acceptable for publication, you may indicate that here to bypass the “Comments to the Author” section, enter your conflict of interest statement in the “Confidential to Editor” section, and submit your "Accept" recommendation.

Reviewer #1: (No Response)

Reviewer #2: All comments have been addressed

2. Is the manuscript technically sound, and do the data support the conclusions?

Reviewer #1: Partly

Reviewer #2: Yes

3. Has the statistical analysis been performed appropriately and rigorously? 

Reviewer #1: Yes

Reviewer #2: Yes

4. Have the authors made all data underlying the findings in their manuscript fully available?

Reviewer #1: Yes

Reviewer #2: Yes

5. Is the manuscript presented in an intelligible fashion and written in standard English?

Reviewer #1: Yes

Reviewer #2: Yes

6. Review Comments to the Author

Reviewer #1: The authors have made a good effort to revise the manuscript. Well done. There are still some things that I suggest you modify, avoiding repetition and clarifying the graphs.

1. Line 365-372.

Repetition of results delete.

Instead you can just say something like:

“Previous studies suggest temperature as well as benthic structure as important factors governing habitat use (51; 52; 53), which might explain the differences in abundance we recorded across atolls”.

2. Line 393-397. Repetition of Results. Delete.

3. Lines 409-413. Delete this repletion of Results. “We detected emigration on three of the investigated sites for hawksbill turtles and on one for greens (see Table S2). At all but one of these four sites, random emigration was observed, indicating a constant rate of association and dissociation of turtles at these sites.”

4. On your graphs try and include a title and units, especially when presenting values.

e.g. Figure 2a. Include a unit

5. Figure 4. Think about the units of this figure. You say multiyear growth rate. This implies the growth of individuals. But I think this figure is about population size growth. So again, think carefully about your figures to avoid confusion.

6. Have a look at figures and decide which you think are clearest and follow them as a template. For example, x-y plots are typically wider than high. Shades of blue are perhaps not the clearest ?

7. Line 133. I think it is still not clear how you identified males V females and I am certain some of your sub-adults will be called “females”. You say:

“… those without a size estimate or clear identification as males based on tail length were recorded as sex and life stage unknown.”

But surely if they were not males, then you would call them females ?

Readers will be suspicious and if they are suspicious here, they might not lose faith with other aspects of your work. So I think it is far better to be honest and simply state something like:

“It is possible that females may have been wrongly identified as sub-adults and vice versa”.

Well done with making tremendous improvements to this manuscript. Graeme Hays

Reviewer #2: The authors have addressed the comments of the reviewers and it is greatly improved. Note, there are some typos and formatting issues in the discussion (minor).

7. PLOS authors have the option to publish the peer review history of their article (what does this mean?). If published, this will include your full peer review and any attached files.

Reviewer #1: No

Reviewer #2: No

---

## [Author Response · Author response to Decision Letter 1]

19 Mar 2023

A response to reviewers comments was included as part of the upload package.

---

## [Editor Report · Decision Letter 2]

21 Mar 2023

A brighter future? Stable and growing sea turtle populations in the Republic of Maldives

PONE-D-22-19155R2

Dear Dr. Hudgins,

We’re pleased to inform you that your manuscript has been judged scientifically suitable for publication and will be formally accepted for publication once it meets all outstanding technical requirements.

Kind regards,

Graeme Hays

Academic Editor

PLOS ONE

Additional Editor Comments (optional):

The authors have made a good effort to revise the manuscript in line with the comments. Thank you to the authors for attending to the comment so thoroughly. I think this manuscript can now be accepted for publication in PloS1. It will make a nice contribution. Graeme Hays
---

## [Editor Report · Acceptance letter]

3 Apr 2023

PONE-D-22-19155R2 

A brighter future? Stable and growing sea turtle populations in the Republic of Maldives 

Dear Dr. Hudgins:

I'm pleased to inform you that your manuscript has been deemed suitable for publication in PLOS ONE. Congratulations! Your manuscript is now with our production department. 

Kind regards, 

on behalf of

Professor Graeme Hays 

Academic Editor

PLOS ONE